# Effects of Obesity and Asthma on Lung Function and Airway Dysanapsis in Adults and Children

**DOI:** 10.3390/jcm9113762

**Published:** 2020-11-22

**Authors:** Ebymar Arismendi, Marina Bantulà, Miguel Perpiñá, César Picado

**Affiliations:** 1Servei de Pneumologia, Hospital Clínic de Barcelona, Universitat de Barcelona, 08036 Barcelona, Spain; earismen@clinic.cat (E.A.); bantula@clinic.ub.es (M.B.); 2Institut d’Investigacions Biomèdiques August Pi i Sunyer (IDIBAPS), 08036 Barcelona, Spain; 3Centro de Investigaciones en Red de Enfermedades Respiratorias (CIBERES), 08036 Barcelona, Spain; 4Neumología y Salud, 46026 Valencia, Spain; perpina.tordera@gmail.com

**Keywords:** asthma, dysanapsis, lung function, overweight, obesity

## Abstract

Obesity increases the risk of developing asthma in children and adults. Obesity is associated with different effects on lung function in children and adults. In adults, obesity has been associated with reduced lung function resulting from a relatively small effect on forced expiratory volume in 1 s (FEV_1_) and forced vital capacity (FVC), with the FEV_1_/FVC ratio remaining unchanged or mildly increased (restrictive pattern). In contrast, in children, obesity is associated with normal or higher FEV_1_ and FVC but a lower FEV_1_/FVC ratio (obstructive pattern). This anomaly has recently been associated with a phenomenon known as dysanapsis which results from a disproportionate growth between lung parenchyma size and airway calibre. The mechanisms that promote disproportionate lung parenchyma growth compared with airways in obese children remain to be elucidated. Obesity and dysanapsis in asthma patients might contribute to asthma morbidity by increasing airway obstruction, airway hyper-reactivity and airway inflammation. Obesity and dysanapsis in asthma patients are associated with increased medication use, more emergency department visits, hospitalizations and systemic corticosteroid burst than patients with normal weight. Dysanapsis may explain the reduced response to asthma medications in obese children. Weight loss results in a significant improvement in lung function, airway reactivity and asthma control. Whether these improvements are associated with the changes in the dysanaptic alteration is as yet unclear.

## 1. Introduction

Obesity and asthma are common diseases in developed countries with their prevalence progressively increasing and affecting people in different proportions all over the world [1,2,3].

Obesity is classically defined as an excess of body weight in relation to height. Body weight regulation results from a complex interaction between genetic, epigenetic, environmental, personal behaviour and socioeconomic factors [2]. However, underlying the simple definition of obesity there is a complex pathophysiological process associated with an increased risk of developing chronic diseases such as type 2 diabetes, cardiovascular diseases, obstructive sleep apnoea and certain cancers. Adipose tissue is an endocrine organ that produces numerous bioactive mediators, such as interleukin (IL) 6 (IL-6), IL-17, IL-1β, IL-12, tumor necrosis factor alpha (TNFα) and leptin, which results in a low degree chronic inflammatory state. Some of these immune changes are also present in asthma patients and are even augmented in obese asthmatics [4,5].

Several longitudinal epidemiological studies have shown that obesity is a major risk factor for asthma in children [6,7,8,9,10,11]. The link between obesity and asthma is not merely mediated by overweight in children because it can start during pregnancy. Maternal obesity and weight gain during the pregnancy are both associated with an increased risk of asthma [12,13]. Similarly, several prospective studies in adults have also found a relationship between obesity and incident asthma [14,15,16].

All in all, maternal obesity, and obesity in both children and adults have contributed to the marked increase in the prevalence of asthma in many countries during the last two decades.

Interestingly, despite excess weight appearing to be a risk factor for asthma in both children and adults, the impact of obesity on lung function differs between children and adults.

## 2. Effects of Obesity on Lung Function in Adults

The first studies aimed at assessing the effects of obesity on lung function were performed in adults and mostly reported that being overweight is associated with reduced lung function [17,18,19,20,21,22,23]. Both forced expiratory volume in 1 s (FEV_1_) and forced vital capacity (FVC) are reduced in obese subjects. As both FEV_1_ and FVC are similarly affected by obesity, the FEV_1_/FVC ratio usually remains unaltered or increases slightly in obese subjects [17,18,19,20,21,22,23]. Abramson et al. [24] found that increase in BMI predicts FEV_1_ decline over a 20-year follow-up in females. The same study reported that FVC decline was related to the increase in body mass index (BMI) in both male and females.

Some studies investigated the effect of obesity on lung volumes and found that obesity weakly affects total lung capacity (TLC) [19,22,25]. TLC is the volume of air contained in the lung after a full inspiration. A decrease in TLC indicates the presence of a restrictive lung function abnormality.

Functional residual capacity (FRC) and expiratory reserve volume (ERV) have also been assessed in some studies [17,19,22,25,26]. FRC represents the equilibrium point of elastic retraction between the lung and the chest wall. FRC results from the sum of residual volume (RV) and ERV. RV represents the amount of air that remains in the lung after a maximal expiratory effort. A decrease in ERV is consistently found in all studies [17,19,22,25,26] in obese subjects, and most of them also reported a similar decrease in FRC [17,19,22,26] in the same subjects. Decrease in FRC and ERV are usually associated with a restrictive lung function abnormality.

The restrictive lung pattern found in obese subjects has been linked to the accumulation of fat in the chest and abdominal cavities, that results in a limited downward excursion of the diaphragm, increase in pleural pressure and significant decrease in the compliance of the entire respiratory system (lung plus chest wall compliance) [27]. These alterations in lung function reduce end-expiratory lung volume, thereby reducing the tethering forces of the parenchyma on the small airways and contributing to increased collapsibility of the peripheral airways which results in increased airway resistance [27,28,29].

Obesity in adults with asthma has been associated with reduced response to asthma medications, resulting in worse disease control, higher risk of hospitalization and lower quality of life than lean asthma patients [4,30,31,32]. Weight loss intervention by caloric restriction [31] or bariatric surgery [33] leads to improvements in asthma outcomes. Reduced systemic and airway inflammation, improvement in the mechanical properties of the lung and of airway reactivity, can account for the observed clinical benefits of weight loss. However, what proportion of the improved asthma outcomes results from reducing the negative effect of obesity on lung function is as yet unclear.

## 3. Effects of Obesity on Lung Function in Children: Role of Dysanapsis

In contrast to adults, some studies in children showed an inconsistent relationship between obesity and lung function and, surprisingly, most others found a positive rather than an inverse association between increased weight and increased lung function [34,35,36]. These observations suggest that obesity during pregnancy and in early years of life can contribute to regulating lung development and growth by mechanisms that remain to be elucidated.

The effects of overweight on lung function in children have been assessed in cross-sectional and cohort studies.

### 3.1. Effects of BMI on Lung Function in Children (Cross-Sectional Studies)

The impact of excess weight and obesity on lung function was the objective of a recent meta-analysis that evaluated various cross-sectional studies in children [36]. The meta-analysis showed that overweight/obese children had slightly higher FEV_1_ values associated with a much higher FVC compared with non-overweight children. The ratio between FEV_1_/FVC (using the fixed cut-off of <0.7 or 0.8), which is commonly used to establish the presence of airflow obstruction was found lower in overweight/obese children (−2.4%, 95%CI: −3.0–1.8%) than in children with the expected normal weight, supporting the notion that overweight/obesity is associated with airflow obstruction.

Another parameter used to measure airway obstruction is the mean forced expiratory flow between 25% and 75% of FVC (FEF_25–75_). The meta-analysis found that the FEF_25–75_ decreased significantly among overweight and obese children, further supporting the negative association between being overweight and airway calibre [36].

### 3.2. Effects of BMI on Lung Function in Children (Cohort Studies)

The temporal relationship between obesity and lung function development and evolution has been examined in some longitudinal studies in children. In the Dutch PIAMA cohort, persistent but not transient high BMI was associated with lower FEV_1_/FVC in children at 12 years of follow-up [34]. Similarly, the Swedish BAMSE cohort analysed the associations between lung function assessed by spirometry and force oscillation technique (FOT) [35]. FOT is an effort-independent technique that allows the assessment of both lung resistance and reactance. Ekström S et al. [35], found that persistent, but not transient, excess weight/obesity between 6 and 16 years was associated with higher lung resistance and reactance, and lower FEV_1_/FVC in girls and boys at 16 years, compared with children who maintain normal weight over the eight years of follow-up. These findings suggest that in children and adolescents, persistent overweight is associated with airway obstruction of the large and small airways. In a recent study, Forno and cols [37] found that overweight and obesity in children were associated with higher FEV_1_ and FVC and low FEV_1_/FVC ratio, compared with non-overweight children in a cross-sectional and follow up study.

A recent meta-analysis from 24 birth cohorts with 25,000 children found that greater birth weight and persistent infant weight gain were both associated with higher FEV_1_ and FVC. Because the increases were greater for FVC than for FEV_1_, infant weight gain was associated with lower FEV_1_/FVC, a lung function finding that supports the presence of overweight-related airway obstruction [38].

### 3.3. Obstructive Lung Function Pattern in Overweight/Obese Children: Role of Dysanapsis

The tendency of overweight/obese children to have higher FEV_1_ and FVC, lower FEV_1_/FVC ratio and lower FEF_25–75_/FVC ratio, than their no-overweight/obese counterparts, has been recently associated with a phenomenon known as dysanapsis [36]. Green and colleagues [39] coined the term “dysanaptic growth” (unequal growth) to describe the concept of the disproportionated growth between lung size and airway calibre. Dysanapsis may explain, at least in part, the wide variations in maximum expiratory flows in healthy subjects with similar lung size. Green and cols [39] postulated that such differences would have an embryological basis reflecting a normal but disproportionate growth within an organ (dysanaptic growth), and that the inequalities in the relationship between airways and parenchyma could influence in the pathogenesis of some respiratory diseases. Somewhat later, J Mead [40] argued that “if persons with large lungs have airways that are not correspondingly larger than those of persons with small lungs, it follows that the ratio of a measurement sensitive to airways size (FEV1) to one sensitive to lung size (FVC), i.e., the FEV_1_/FVC ratio, should vary reciprocally with lung size”. In Mead’s other words: “if lung and airway size changed together, the FEV_1_/FVC ratio would be the same for large and small lungs, i.e., for persons with large and small vital capacity” [40]. He proposed to use a “dysanaptic index” (also known as dysanaptic ratio) to express the interindividual discrepancy between parenchyma and airway size [40].

The dysanaptic index can be calculated using the equation FEF_50_/FVC × Pst(l)50%. FVC is a measure of lung size, FEF_50_ (forced expiratory flow at 50% of FVC) is an indirect measure of airway size, while Pst(l)50% is the elastic recoil pressure of the lung at 50% of the FVC (pressure that generates flow) [40]. Pst(l)50% is estimated using the equation proposed by Turner et al. [41] Pst(l)50% = −0.056 × age + 6.3038 based on age. A lower index indicates more marked dysanapsis and predicts expiratory flow limitation [42].

The measure of the dysanaptic index was later simplified by using the FEF_25–75_/FVC ratio as a surrogate [43].

Mead et al. [40] calculated that the airway calibre for males was 17% higher than that of women. These differences were later confirmed using acoustic reflectance [44,45] and high-resolution computed tomography techniques [46]. Differences in airway calibre between sexes may explain why women develop a greater resistance of the airways during exercise, which may result in a higher metabolic cost of breathing and reduced exercise capacity of women compared to men [46,47].

The hypothesis that establishes a relationship between obesity and dysanapsis is based on the results of epidemiological studies that show an association between weight gain during pregnancy [12,13] and the first years of life with an alteration in the lung development that consists in the increase of the lung size-measured by FVC- associated with a smaller growth and reduced calibre of the airways- assessed by FEV1, which results in a decreased FEV1/FVC ratio indicating the presence of expiratory flow limitation [36,37,38].

These observations suggest that the excessive increase in adipose tissue in obese patients can regulate the growth and development of the lung structures [36,37]. However, the physiopathologic mechanisms involved in the dysanaptic growth of lungs in obese children remain to be fully elucidated. Some observations suggest that the increased release of proinflammatory substances produced by the hypertrophied adipose tissue in obese subjects can contribute to the abnormal growth of lung [48,49,50,51,52]. Leptin, which is produced in large amount by the adipose tissue from obese patients, appears to be involved in embryonic lung growth and maturation, promoting neonatal lung development and surfactant A production [15,48,49]. Polymorphisms in leptin and its receptor have been associated with lung function alterations, a finding that further supports the potential regulatory role of leptin in lung development [50,51]. Furthermore, leptin can increase the release of cytokines produced by macrophages of the adipose tissue, such as IL-6, IL-12 and TNFα, which exert regulatory effects on lung development [52]. Overall, these observations suggest that leptin and increased production of other yet unknown bioproducts released by the adipose tissue may contribute to dysanapsis in overweight/obese children.

## 4. Obesity and Asthma: Combined Effects on Lung Function and Airway Dysanapsis

Many conditions, other than obesity, have been found with dysanapsis such as diving (professional divers) [53], vitamin D deficiency [54] and chronic hypoxia living at high altitude [55].

Asthma initiated in childhood has also been found associated with increased lung volumes (FVC) in older [56,57] and more recent studies [58], a finding that supports the hypothesis that asthma initiated at early years may play a regulatory role in lung growth and development. Jones and colleagues [58] performed a study in 188 children, in which 74 (39%) children were healthy and 111 (61%) were asthmatics, while 72 (38%) of the children with asthma were overweight or obese. Children with excess weight had a higher FVC and FEV_1_ and lower FEV_1_/FVC than healthy controls. Moreover, overweight/obese children with asthma had a higher FVC and lower FEV_1_/FVC compared with normal weight asthmatics. The authors suggested that both asthma and excess of weight are associated with dysanapsis. However, the contribution of excess of weight to dysanapsis was substantially larger than that of asthma, and there was no interaction between excess weight and asthma on lung function. The mechanisms potentially involved in the development of dysanapsis in patients with asthma initiated in early years remain to be clarified.

### Dysanapsis and Asthma Morbidity

A recent study examined the relationship between dysanapsis and clinical outcomes in children with asthma and found that asthmatic children with dysanapsis used more medications to treat asthma, required more rescue medication and reported more emergency department visits, hospitalizations and systemic corticosteroid burst than children with normal weight [37]. Forno and colleagues [37], speculate that dysanapsis may, at least partly, explain the reported association of obesity with worse asthma control and reduced response to asthma medications.

The mechanisms linking dysanapsis with worse asthma control and severity are yet unclear. Airflow obstruction due to asthma associated with airway obstruction caused by dysanapsis may contribute, via an additive or synergistic effect, to increase the airflow limitation in obese children with asthma [59].

Dysanapsis has also been associated with airway hyperreactivity. Various epidemiological studies have confirmed these findings. Tager and cols [43], in a group of children and adolescents, demonstrated that the lower the baseline values of the dysanaptic ratio FEF_25–75%_/FVC the higher the bronchoconstrictor response to eucapneic hyperventilation with cold air, especially in those subjects with a history of wheezing or with an asthma diagnosis. Litonjua and cols [60] found a significant association between the FEF_25–75%_/FVC ratio with response to methacholine after adjusting for age, sex, atopy, immunoglobulin E (IgE) serum levels, blood eosinophil numbers and baseline FEV_1_. Urrutia and cols [61] also reported a significant association between the ratio FEF_25–75%_/FVC and methacholine dose needed to cause a 20% drop in the basal FEV_1_ after adjusting for age, sex, atopy, IgE and respiratory symptoms. Parker and cols [62] confirmed the determinant role played by the dysanaptic ratio in airway reactivity and sensitivity to methacholine.

Taken together, these findings suggest that dysanapsis might contribute to asthma morbidity in children by increasing both airway obstruction and airway hyperreactivity.

As mentioned earlier, weight loss in obese adults with asthma may have therapeutic effects; however, weight loss by hypocaloric diets in obese children with asthma has only been investigated in a few studies. Studies in adults found that weight loss was associated with better asthma control, reduced asthma exacerbations, reduced exercise-induced bronchoconstriction and improved quality of life [63,64,65].

Whether the low dysanapsis ratio found in obese children remains unaltered or improves when children enter adulthood is a yet unclear. Similarly, is not known if weight loss can partially or totally reverse the dysanaptic alteration of lung function in obese children.

## 5. Conclusions

Overweight and obesity increase the risk of developing asthma in children and adults. Obesity is associated with a different type of effect on lung function in children and adults. In adults, obesity has been associated with reduced lung function (restrictive pattern), while in children, obesity is associated with an obstructive pattern resulting from disproportionated growth between lung parenchyma size and airway calibre, a phenomenon known as dysanapsis. The mechanisms that promote disproportional lung parenchyma growth in comparison to airways in obese children are yet unknown. Asthma in children associated with obesity and dysanapsis result in increased medication use, more emergency department visits, hospitalizations and systemic corticosteroid burst than in children with asthma and normal weight. Whether these clinical improvements are associated with the changes in the dysanaptic alteration remains to be studied.

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
