# Peer review of "Effects of Obesity and Asthma on Lung Function and Airway Dysanapsis in Adults and Children"

_jcm, 2020, doi:10.3390/jcm9113762_

Round 1
Reviewer 1 Report
This review discusses the effect of obesity and asthma on lung function and airway dysanapsis. The authors first compare the effects of obesity on lung function in children and adults. Then they discuss the association between obstructive lung function pattern and lung dysanapsis in overweight/obese children, and the combined effect of asthma and obesity on lung function and dysapnasis in children.
While the authors review the literature in detail, the text is confusing at some places, particularly when the authors report findings from the literature. I recommend authors to revise the text for clarity, and add author’s thoughts/interpretation for the findings discussed (a short summary). There are multiple single sentence paragraphs throughout the text. I suggest authors to carefully review and combine those associated with common discussion points.

Reviewer 2 Report
Note that the last sentence of the manuscript "This section is not mandatory, but may be added if there are patents resulting from the work reported in this manuscript" needs to be deleted.

Reviewer 3 Report
The authors review current knowledge of obesity and dysanapsis in asthma. This manuscript is well-written and worthwhile reading. I think that this review article will contribute the understanding of pathophysiology of asthma.
1. It is better to describe details of how dysanapsis is associated with higher FEV1 and FVC and lower FEV1/FVC. The association between dysanapsis and lung function (FEV1 and FVC) and its relevance to obesity is not fully explained.
2. Line 179, Please describe why the authors concluded that both asthma and overweight are associated with dysnapsis.
3. Line 173, I think "then" should be changed to "them".
4. Please remove the last sentence of the conclusion section.
Round 2
Reviewer 1 Report
Effects of Obesity and Asthma on Lung Function and Airway Dysanapsis in Adults and Children
The authors have addressed most of my concerns. I have a few minor comments and suggestions(mainly related to grammar), which appear in bold text below. I also recommend the authors using active voice wherever possible and avoiding single sentence paragraphs.
Specific Comments:
Line 12: ...with different effectson lung function...
Line 21: Obesity and dysanapsis in asthma patients...
Line 54-55: I missed this in the earlier version. Do you mean both FEV1 and FVC are reduced? It is unclear what you mean by both lower FEV and lower FVC.
Line 82: ...on lung function is yet unclear...
Line 129: to describe the concept of disproportionategrowth... [As used on line 132]
Line 131: expiratory flows in healthy children?
Line 137: ... size (FEV1) to the one sensitive to lung size... Line 151: ... and high resolution computed tomography techniques.
Line 156: Very long sentences can be confusing. I recommend authors to revise long sentences for better readability.
Line 166: Please remove one of the two periods at the end of this text.
Line 167: ... tissue in obese patients...
Line 169: ...such as diving (professional divers)... [Colon after ‘such as’ is not required, Suggestion to remove it]
Line 174:... other yet unknown byproducts released by adipose tissue may contribute...
Line 179: ...has been found to be associated with...
Line 182: ...a study in 188 children in which 74 (39%) children were healthy and 111 (61%) children were asthmatics, while 72 (38%) of the children with asthma were overweight or obese... Line 186: The authors suggested that both asthma and excess weight are associated with dysanapsis.
Line 187: However, the contribution of excess of weight to dysanapsis was substantially larger than that to asthma, and there was no interaction between excess weight and asthma on lung function.
Line 194: found that asthmatic children with dysanapsis used more [Please remove comma between dysanapsis and used]
Line 199: ...severity are yet unclear.
Line201-202: ... the airflow limitation in children with asthma and obesity.” Do you mean obese children with asthma? If so, please revise.
Line 206-207: Suggested active voice instead of passive voice: Various epidemiological studies have confirmed these findings. Line 210:2011: ...a significant association between FEF25-75%/FVC ratio andmethacholine dose needed to...in the baselineFEV1...
Line 214: Please move this single sentence to the preceding paragraph (lines 203-213) as it is the concluding text for this paragraph. Line 208: ...association between the ratio FEF25-75%/FVCandresponse to methacholine after ..
Line 222: ...enter adulthood is yet unclearLine 226: ...with different effects...?
Line 228: disproportionate?
Line 230-231: ...are yet unknown.
Line 231: Do you mean: In children with asthma, obesity and dysanapsis result in increased medication use...
If you wish to leave the text on line 231 as it is, please revise as: ...associated with asthma in children result in...

Author Response
Reviewer (R). The authors have addressed most of my concerns. I have a few minor comments and suggestions(mainly related to grammar), which appear in bold text below. I also recommend the authors using active voice wherever possible and avoiding single sentence paragraphs.
Authors (A). Thank you very much again for your suggestions and comments. All of them have helped us to improve the quality of both the writing and the adequate expression of the scientific and clinical concepts discussed in the manuscript
Specific Comments:
Line 12: ...with different effects on lung function...
Line 21: Obesity and dysanapsis in asthma patients...
Line 54-55: I missed this in the earlier version. Do you mean both FEV1 and FVC are reduced? It is unclear what you mean by both lower FEV and lower FVC.
Line 82: ...on lung function is yet unclear...
Line 129: to describe the concept of disproportionategrowth... [As used on line 132]
Line 131: expiratory flows in healthy children?
Line 137: ... size (FEV1) to the one sensitive to lung size... Line 151: ... and high resolution computed tomography techniques.Line 156: Very long sentences can be confusing. I recommend authors to revise long sentences for better readability.
Line 166: Please remove one of the two periods at the end of this text.
Line 167: ... tissue in obese patients...
Line 169: ...such as diving (professional divers)... [Colon after ‘such as’ is not required, Suggestion to remove it]
Line 174:... other yet unknown byproducts released by adipose tissue may contribute...
Line 179: ...has been found to be associated with...
Line 182: ...a study in 188 children in which 74 (39%) children were healthy and 111 (61%) children were asthmatics, while 72 (38%) of the children with asthma were overweight or obese... Line 186: The authors suggested that both asthma and excess weight are associated with dysanapsis.
Line 187: However, the contribution of excess of weight to dysanapsis was substantially larger than that to asthma, and there was no interaction between excess weight and asthma on lung function.
Line 194: found that asthmatic children with dysanapsis used more [Please remove comma between dysanapsis and used]
Line 199: ...severity are yet unclear.
Line201-202: ... the airflow limitation in children with asthma and obesity.” Do you mean obese children with asthma? If so, please revise.
Line 206-207: Suggested active voice instead of passive voice: Various epidemiological studies have confirmed these findings. Line 210:2011: ...a significant association between FEF25-75%/FVC ratio andmethacholine dose needed to...in the baselineFEV1...
Line 214: Please move this single sentence to the preceding paragraph (lines 203-213) as it is the concluding text for this paragraph. Line 208: ...association between the ratio FEF25-75%/FVCandresponse to methacholine after ..
Line 222: ...enter adulthood is yet unclearLine 226: ...with different effects...?
Line 228: disproportionate?
Line 230-231: ...are yet unknown.
Line 231: Do you mean: In children with asthma, obesity and dysanapsis result in increased medication use...
If you wish to leave the text on line 231 as it is, please revise as: ...associated with asthma in children result in...
Authors (A). We have made the suggested changes. We have cleared up some confusing statements and fixed the errors noted in the manuscript.
Reviewer 2 Report
Title: the title now emphasizes that the paper is about obesity and asthma and their effects on lung function and dysanapsis in adults and children. I agree that the paper describes the effects of obesity on lung function and dysanapsis in adults and children, but not how asthma affects dysanapsis in adults and children. I think the original title was better.
p.1, L21: “obese” should be “obesity”
p.2, L54: the change is incorrect – “both lower … FEV1 and lower… FVC are reduced in obese subject” is redundant; it should either read “Both … FEV1 and … FVC are reduced…” or read as the original “Both lower …FEV1… and …FVC have been reported…”
p.2, L90: to be consistent with the changes made elsewhere, “overweight” should be changed to “obesity”
p.3, L102: the definition of FEF 25-75 is still incorrect: it is the average forced expiratory flow between 25% and 75% of the FVC. The authors have changed the definition to suggest that it is the ratio FEF25-75 divided by FVC (which is later referred to as the dysanaptic index as stated on pl. 4, L148)
p.3, L129: should be “disproportional”
p.4, L152: This is a long and awkward sentence. Suggest rewording:
"Differences in airway calibre between sexes may explain why women develop greater airway resistance during exercise, which may result in a higher metabolic cost of breathing and reduced exercise capacity compared to men. "
p.4, L156-166; this is a new set of ideas which requires references – what are the “epidemiological studies” referred to here; and what is the evidence for “excessive increase in adipose tissue… can regulate the growth and development of the lung structures”?
p. 4, L170: should be “further supports…”
p.4, L172: should be “bioproducts released…”
Author Response
Thank you very much for helping to improve the writing and expression of both the clinical concepts and the parameters used in the study of lung function.
Reviewer (R): Title: the title now emphasizes that the paper is about obesity and asthma and their effects on lung function and dysanapsis in adults and children. I agree that the paper describes the effects of obesity on lung function and dysanapsis in adults and children, but not how asthma affects dysanapsis in adults and children. I think the original title was better.
Authors (A): Finding a sentence for the title of a scientific article is usually not easy when a complex relationship of several concepts must be condensed into a few words. The original title was questioned by one of the reviewers of the manuscript who considered it "confusing". We agreed that the title of the article was uncomfortable to read. Following the reviewer's proposal, we modify the title. The current one does not completely satisfy us as it does not fully reflect the message of the manuscript, but it is also true that it is an open title, in the sense that what is not expressed is not excluded either.
Reviewer (R): p.1, L21: “obese” should be “obesity”
p.2, L54: the change is incorrect – “both lower … FEV1 and lower… FVC are reduced in obese subject” is redundant; it should either read “Both … FEV1 and … FVC are reduced…” or read as the original “Both lower …FEV1… and …FVC have been reported…”
p.2, L90: to be consistent with the changes made elsewhere, “overweight” should be changed to “obesity”
Authors (A): Errors have been corrected and suggestions included in the revised manuscript
Reviewer (R): p.3, L102: the definition of FEF 25-75 is still incorrect: it is the average forced expiratory flow between 25% and 75% of the FVC. The authors have changed the definition to suggest that it is the ratio FEF25-75 divided by FVC (which is later referred to as the dysanaptic index as stated on pl. 4, L148)
Authors (A): We apologize for making the mistake again despite your warning in the previous review. We have corrected it.
Reviewer (R): p.3, L129: should be “disproportional”
p.4, L152: This is a long and awkward sentence. Suggest rewording:
"Differences in airway calibre between sexes may explain why women develop greater airway resistance during exercise, which may result in a higher metabolic cost of breathing and reduced exercise capacity compared to men. "
Authors (A): The spelling mistake has been corrected and the sentence modified according to your suggestion.
Reviewer (R): p.4, L156-166; this is a new set of ideas which requires references – what are the “epidemiological studies” referred to here; and what is the evidence for “excessive increase in adipose tissue… can regulate the growth and development of the lung structures”?
Authors (A): There is no evidence. What is established is a hypothesis made from observations in epidemiological studies. As these show that obesity is associated with dysanapsis, it seems logical to think that obesity alters the proportional growth of the lung (parenchyma) and airways. The mechanism is also unknown, although there are a number of observations that suggest that some of the metabolic disturbances associated with obesity may influence the harmonious (proportionate) development of the lung. References are indicated in the manuscript.
Reviewer (R): p. 4, L170: should be “further supports…”
p.4, L172: should be “bioproducts released…”
Authors (A): Spelling mistakes corrected.
Reviewer 3 Report
The authors responded to all of the suggestions properly. Especially, the current knowledge about associations between lung function and obesity and dysanapsis has been clearly explained. The manuscript is improved and worthwhile reading.
Author Response
Thank you for your kind comments and help.